# Thermal Performance of Multifunctional Facade Solution Containing Phase Change Materials: Experimental and Numerical Analysis

**DOI:** 10.3390/polym15132971

**Published:** 2023-07-07

**Authors:** C. Amaral, F. Gomez, M. Moreira, T. Silva, R. Vicente

**Affiliations:** 1TEMA—Centre for Mechanical Technology and Automation, Department of Mechanical Engineering, University of Aveiro, 3810-193 Aveiro, Portugal; 2LASI—Intelligent Systems Associate Laboratory, 4800-058 Guimarães, Portugal; 3AMS—Advanced Material Simulation, C/Asturias n°3, 48015 Bilbao, Spain; 4RISCO—Research Center for Risks and Sustainability in Construction, Civil Engineering Department, University of Aveiro, 3810-193 Aveiro, Portugal; romvic@ua.pt

**Keywords:** multifunctional facade panel, phase change material (PCM), hotbox testing, thermal transmittance, numerical simulations

## Abstract

This work focuses on the development and analysis of a new multifunctional facade panel incorporating PCM in foam layers. The thermal performance was analysed recurring to a hotbox heat flux meter method to determine the thermal transmittance (U-value) and the main findings are presented. The experimental setup was based on the steady-state approach, using climatic chambers, assuring a stable thermal environment. Even small fractions of PCM achieved a small reduction in thermal amplitude. Numerical simulations using Ansys Fluent were developed to evaluate the performance of PCM use over a wide range of temperature boundary conditions and operating modes. These numerical models were calibrated and validated using the results of experimental tests, achieving a correlation factor of 0.9674, and, thus, accurately representing a real-world scenario. The decrement factor (f) was used to analyse the data. It was identified that the efficiency of the panel and size of the optimum region increased with the PCM fraction growth. The results showed the significant potential of the multi-layered panel, with the thermal regulator effect of the PCM incorporated, on indoor space temperature so as to reach good thermal comfort levels. The efficiency of the panel can be improved by nearly 50% depending on the input boundary conditions. The efficiency of the panel and the size of the optimum region increase with growth in the PCM fraction. The simulated behaviour was at an optimum when the input mean temperature was 20 °C for a room temperature of between 18–20 °C.

## 1. Introduction

Reduction in the energy consumption of buildings has always been on the EU agenda as a flagship topic, bringing together the research community and industry in joint efforts to reduce energy dependency, as a way to decrease resource consumption (water, energy, raw materials) to more environmentally friendly levels [1]. One of the highest energy-consuming sectors is housing, which consumes close to 30% of total energy consumption, according to the International Energy Agency (IEA) [2,3,4]. This energy consumption is also reflected in a large amount of carbon dioxide emissions [5,6]. Building fabric, namely the external envelopes, as in the case of facades, play a relevant role in reducing the energy demands of buildings. There are wo essential requirements in regard to facades: initially, they act as barriers between a building′s interior and the external environment, providing a liveable space for inhabitants, and, secondly, there is the aesthetical component of the image of the building, responsible for its desirability. 

High-performance sustainable facades are set as exterior enclosures designed to minimize energy usage while ensuring a comfortable indoor environment, fostering a healthy and productive space for building occupants. These facades are not mere barriers between interior and exterior spaces. They are building systems, capable of responding to conditions imposed by the external environment and effectively reducing energy consumption in buildings. Thermal transmittance (U-value) is one of the most used parameters to assess the energy efficiency of building elements. The definition of thermal transmittance, also known as the heat transfer coefficient or U-value (W/(m^2^.K)), is the heat flow rate divided by the wall area and by the difference in temperature between the two sides [7,8]. 

One of the ways to better regulate indoor temperature in a passive manner is through the use of thermal energy storage (TES) materials, such as phase change materials (PCMs). The potential of these materials is considerable because they have high latent heat, making them very efficient in storing energy [9,10], which can help to stabilize indoor temperatures during the daily cycle. 

In recent years, several research works on the incorporation of PCMs in opaque building solutions, radiant floors, and glazing have been conducted using different encapsulation techniques and methods. The most commonly used type of PCM encapsulation are micro- and macro-encapsulation, and for the latter type of encapsulation there are different forms and shapes [11,12,13,14]. Other relevant factors for the correct use of PCMs in building facades are the melting/solidification temperatures, positioning, latent heat capacity and average weather climatic conditions [15,16,17,18,19,20]. 

Sovetova et al. [16] presented a numerical analysis to evaluate the use of a microencapsulated PCM layer located between the outer layer (cement plaster and ceramic tile) and concrete layer on the façade of a building in a hot desert region. They concluded that the energy consumption reduction (ECR) could reach 34.26%, and an economic analysis showed that the investment was feasible. In other studies [17,18], an aluminium tubular system with macro-encapsulated PCM was analysed and integrated into a wall. The authors concluded that there was a thermal amplitude reduction in all walls and the roof, ranging from 40.67% to 59.79%. Besides this, the authors obtained a reduction of about 7% to 9% in the internal temperature of the room, as well as a reduction in the ambient thermal load of about 38% [18]. In addition, the indoor peak temperature improved from 0.2 °C to 4.3 °C.

Bahrar et al. [21] performed a multiscale experimental characterisation of textile-reinforced concrete panels with micro-encapsulated PCM using the hotbox method. They also developed a numerical model to accurately reproduce the thermal performance of the building envelope. The authors concluded that the higher the amount of PCM particles, the lower the thermal conductivity of the studied specimens. Moreover, experiments under real weather conditions showed a reduction of 1.7 °C of the peak internal surface temperature. 

Li et al. [22] assessed thermal, economic and environmental analyses of PCM-embedded walls in rural residences in northeast China, using EnergyPlus^TM^. Regarding the PCM, the studied parameters were the PCM layer position, the PCM wall orientation, and the PCM melting point. The results showed an energy saving of 12.9% by a PCM-filled wall near the interior surface. Compared with the baseline case, the PCM-filled wall in the south facade decreased the heating load by 12.8%. The optimum PCM melting temperature was 16 °C for an interior temperature of 18 °C. In this case, the carbon footprints were reduced by 52.7 kg.m^−2^ when using the appropriate PCM wall. 

Liu et al. [23] tested the thermal and optical behaviours of a non-ventilated multilayer glazing facade filled with PCM. This study was carried out for cold climate conditions and the authors used experimental data to validate the numerical model. The results showed that PCM thickness was the most important parameter in the thermal and optical performances of multi-layer glazing façades. An increase in PCM thickness resulted in lower heat loss from the glazing façade and also increased the time delay and the internal surface temperature. However, it was recommended that the thickness of the PCM layer not exceed 20 mm due to significant decrease in solar transmission.

There are many other relevant studies focusing on PCM applications in buildings in the bibliography. In these studies, information regarding the use, applications, strategies, new products and challenges are presented [24,25,26].

The multifunctional panel developed in the scope of this research has many interesting features, ranging from the use of PCM, recycled materials, and innovative material processing techniques to the auto-cleansing function of the exterior surface. This multifunctional panel can be used in new buildings, as well as in building rehabilitation, making its adoption possible in a broader spectrum of use. Focusing on the thermal performance of the panel, an experimental testing campaign was carried out using the hotbox method, which is one of the most common protocols for building component and solution characterization [27]. This method aims at determining the dynamic U-value as well as the temperature regulation effect from the charging and discharging of the incorporated PCM. Ultimately, the objective of this research was to create a novel numerical model capable of predicting the thermal performance of a multifunctional facade with PCM under different operating conditions. This enables the possibility of characterizing a building’s facade at the design stage and determining its benefits and advantages over normal building construction solutions, to help in the continuous improvement and development of more sustainable buildings.

## 2. Experimental

### 2.1. Materials and Description of the Panel

The multifunctional facade panel is constituted of the following layers, each providing a different functionality within the final multilayer panel configuration. Figure 1a represents the scheme of the multifunctional panel and the position of the corresponding layers. Figure 1b illustrates the overall dimensions of the multifunctional panel, and Figure 1c shows the inner and outer panel sides. The properties of the various materials can be consulted in Table 1. The pristine PCM was synthesized according to Amaral et al. [28] and Table 2 lists its principal properties. More specifically, the multifunctional facade panel is constituted of the following layers with specific functionalities: 

The anchoring layer is formed by three separate polyurethane layers. The soft polyurethane (PU) foam layer is compressed against the outer or existing back wall, covering defects and protrusions. The hard PU foam layer supports thermal insulation and provides mechanical stability to the panel. These two layers incorporate a low percentage of PCM in weight (1.8 wt%). Finally, an inner soft PU foam layer without PCM protects the adjacent insulation layer (aero clay);

The main insulation layer of the façade provides thermal and acoustic insulation (clay/silica aerogel layer);

The durable layer protects the inside layers from humidity, provides mechanical properties and fire protection for the components (geopolymer layer);

The external layer provides flame retardancy and mechanical stability (fibre-reinforced polymer layer);

The intumescent layer provides fire protection and anti-corrosion properties (intumescent fire layer);

The surface coating provides the final aesthetics of the panel and photocatalytic properties (paint coating).

### 2.2. Hotbox Method

The hotbox method is based on the steady-state method and involves a relatively stable thermal environment using a simple setup [29,30,31]. Each hotbox configuration has two closed chambers with controlled temperature and relative humidity conditions. One of the chambers is considered a cold chamber (with a low and constant temperature) and the other is the metering chamber or warm chamber (where the required temperatures are imposed). Between the two chambers a mounting ring is fixed (Figure 1b), with the panel specimen to be tested [31,32,33]. The detailed procedure, schemes, and the materials are described in Amaral et al. [34,35].

The dimensions of the panel for thermal properties evaluation were 800 × 650 mm (height × width) with a thickness of 122.65 mm (see Figure 1a,b). These dimensions were limited by the hotbox mounting ring structure geometry (maximum internal dimensions were 800 × 650 × 390 mm). The climatic chambers and specimen surface were monitored using thermocouples and heat flux meters on the panel specimen, as shown in Figure 2.

Figure 2 shows the sensor positions and quantities on each specimen surface (exterior and inner sides). Eight thermocouples of type-T and two heat flux meters were placed on the external surface of the specimen (warm chamber side). The other eight thermocouples of type-T were fixed on the internal surface of the specimen to measure the internal chamber surface temperatures, as well as heat flux meters placed in the same but opposite positions. In addition, multiple PT100 probes were used in this experimental campaign: five PT100 probes positioned 150 mm from the specimen (cold and warm chambers), one PT100 probe placed in the middle of the chamber (cold chamber side), and six PT100 probes positioned oppositely in the same relative positions (warm chamber).

### 2.3. Experimental Results

#### 2.3.1. Temperature Amplitude

The equivalent thermal conductivity (*λ*) of the whole panel is calculated according to Fourier′s Law, using the following equation:(1)$$\lambda =\frac{q\times s}{\Delta T}$$
where *q* is the measured horizontal heat flux through the specimen in steady state in W/m²; *s* is the thickness of the specimen in m; Δ*T* is the difference between the specimen surface temperatures in °C. The global thermal transmittance can be calculated by:(2)$$U=\frac{\lambda }{s}\left[W/\left(m^{2}.K\right)\right]$$

To measure the thermal transmittance, the temperatures of the chambers were separately set to the following pair temperatures in °C: (2;12), (4;14), (6;16), (…), (30;40), where each pair temperature meant: (cold chamber temperature; warm chamber temperature).

Each step was for 6 h at each temperature difference, summing up to a total amount of 15 steps of measurements.

Figure 3 shows the experimental values determined for the thermal conductivity and thermal transmittance versus the mean temperature of the surface temperatures of both sides for the multifunctional facade panel tested. A fairly good relationship was observed between the thermal conductivity and the thermal transmittance and the increasing mean surface temperature of the specimen. The square correlation coefficient (R^2^) obtained was 0.942, which corroborated good accuracy of the linear fitting. The thermal conductivity values obtained were between 0.0326 and 0.0609 W/m.K and the thermal transmittance values obtained were between 0.2719 and 0.5077 W/m^2^.K for a mean temperature ranging from 12.16 °C to 39.92 °C. During the PCM phase change transition stage a slightly decreasing trend with increasing mean temperature of the specimen was registered. It was found that, as the solid-liquid phase change temperature range of the PU foam with PCM was about 20–23 °C, part of the PU foam would experience phase change when the temperature rose from 15 °C to 25 °C. Therefore, the effect on the thermal conductivity and thermal transmittance profiles with the incorporation of the PCM was not visible, acknowledging that the percentage of PCM incorporated was very low (1.8% in weight), However this feature was numerically assessed with a calibrated model, described in Section 3.

#### 2.3.2. Thermal Amplitude Results

To assess the thermal amplitude reduction between the simulated exterior conditions and the indoor environment (warm and cold chambers), a series of measurements over eight daily cycles were carried out. In this study, the temperature of the cold chamber was defined to free float, while the temperature of the warm chamber followed the presented temperature profile that ranged from 12 °C to 52 °C. These temperature conditions were chosen to represent a typical Mediterranean climate context. Figure 4 shows the experimental results, depicting the temperature profiles of the chambers and specimen surfaces. The air and surface temperature results in the warm chamber were similar. However, a slight difference was observed between the temperatures in the cold chamber, which could be attributed to the free-floating temperature conditions applied to this chamber.

The cold chamber profile ranged between 18 °C and 30 °C for an imposed temperature of 12 °C to 52 °C in the warmer chamber. Comparing the exterior (warm chamber) and the indoor (cold chamber) the thermal amplitude reduction was very significant, specifically for the maximum and minimum peak temperatures in these specific operating conditions. 

## 3. Numerical Models

In order to access the thermal performance of the studied panel, several numerical models were developed, and simulations were performed. Experimental data from the previous sections were used. A geometrical model of the experimental setup was used to validate and calibrate the numerical model, as discussed in the next section.

### 3.1. Numerical Definitions

Recurring to Ansys Workbench v14, a 2D finite element model was developed to simulate the experimental transient thermal testing conditions. A 2D model was used instead of a 3D complex model to reduce the computational requirements and complexity of the model. A refined mesh of 15,3219 elements (elements size of 0.5 mm, and a refinement control applied in the indoor boundary in contact with the cold and metering chamber) was used (Figure 5).

To reduce necessary computational time and requirements, the geometry symmetry, specimen composition, and boundary conditions were defined to allow a smaller middle cross-section model of booth chambers and the mounting ring where the specimen wass mounted (Figure 6). 

The multifunctional facade panel is located at the centre of the model between the two chambers (Figure 1a). A surface divides the cold chamber into two parts to represent the PT100 probe positions at a distance of 150 mm from the panel (“PT100”). The chamber is composed of three regions: the cold chamber, the mounting ring, and the warm chamber (Figure 6). The chamber walls are composed of three layers: inner steel (1.5 mm), rockwool insulation (125 mm), and external zinc with protection (1.5 mm). The boundary conditions introduced in the problem, as shown in Figure 6, were the following: (i) the symmetry plane at the bottom (“Symmetry”), (ii) the uniform temperature at the external boundary (“Room temperature”), and (iii) the warm chamber (“Input temperature”). The numerical calculations were carried out considering the energy equations and the solidification and melting models. 

The facade panel properties were listed in Figure 1 and the material properties of the chamber walls are shown in Table 3 [34]. 

The specific heat of the soft foam layer with PCM and the hard foam layer with PCM, designated as *c_mix_*, were calculated considering the PCM mass fraction, *f_w_*.
(3)$$c_{mix}=\left(1-f_{w}\right)c_{foam}+f_{w}c_{PCM}$$
where *c_foam_* is the specific heat of the base layer and *c_PCM_* is the specific heat of the PCM. The fusion temperature of the PCM is 21–22 °C. The specific heat of the foam layer with PCM during the phase change is equal to:(4)$$c_{mix}=\left(1-f_{w}\right)c_{foam}+f_{w}c_{PCM}+f_{w}L_{v}$$

*L_v_* is the latent heat phase change of PCM. The internal air is modelled as a fluid in laminar regime.

### 3.2. Numerical Validation with Experimental Results

The inputs of the model was the room temperature variation (around the chambers in laboratory conditions) and the experimental temperature curve versus time of the warm chamber. The imposed profile temperature ranged between 12 °C and 52 °C, with a period equal to one day cycle. A total of eight days was considered for the numerical simulation, and a 5 min timestep was defined for the numerical calculations. For each timestep, a maximum of 300 iterations were considered, and the numerical model convergence was found when all residuals’ values were attained. The residual values were set to 1 × 10^−5^ (momentum, energy, and velocity). The room temperature was 18 °C ± 2 °C. The input temperature was taken from the experimental data recorded in the warm chamber, and the room temperature was the reference measured laboratory temperature of the chambers. Figure 7 shows an example of the temperature contours for the timestep when the imposed input temperature was maximum (52 °C = 325.15 K).

Analysing the average temperatures of the cold chamber taken from the PT100 probes or, in the case of the numerical model, the equivalent positioning, the numerical and experimental temperature profiles showed a good fit (see Figure 8).

To assess the accuracy of the numerical model two criteria were used: the goodness-of-fit (GOF) and the correlation factor R^2^.

#### 3.2.1. Statistical Indices

The GOF [36] indicator is a dimensionless index which allowed us to evaluate the calculated results with the measured results. The methodology and equations to calculate GOF can be consulted in [34]. Smaller values of the GOF criteria represent the parameters that provided a closer match between measured data and simulated results. 

Table 4 presents the calculated statistical indices. For each index, the results that presented the best agreement between the experimental and numerical data are presented below:

A GOF value lower than 11% is recommended for trial agreement, so the overall results were acceptable [36].

#### 3.2.2. Correlation Factor, R^2^

An alternative criterion to assess the agreement between the experimental and numerical data was the average of the correlation factor, R^2^. Figure 9 presents a scatter plot between the experimental specimen data and numerical results of the model in the indoor—cold chamber.

Comparing the numerical and experimental data, a correlation factor of 0.9674 was found for the best/optimized model.

### 3.3. Numerical Results

Once the model was validated with experimental data, a parametric calculation was performed to evaluate and analyse the function, performance, and optimum working conditions of the PCM multifunctional facade panel. The input variables defined were the following: (i) the mass fraction of PCM, (ii) the warm chamber mean temperature, (iii) the warm chamber amplitude temperature, and (iv) the room temperature. These variables could range as follows:

PCM mass fraction: 0%, 1.8%, 5% and 10%;

External mean temperature T_mean_: 10 °C, 15 °C, 20 °C, 25 °C, 30 °C and 35 °C;

Temperature amplitude, T_amp_: 1 °C, 5 °C, 10 °C, 15 °C and 20 °C;

Room temperature: 18 °C and 20 °C.

Combining the four variables there were 240 combinations to be studied, this is, to be simulated. For each of these combinations, the decrement factor, *f*, was defined as the ratio between the temperature amplitude of the output with PCM relative to the temperature amplitude without PCM and could vary between 0 and 1:(5)$$f=\frac{\Delta T_{Panel\;with\;PCM}}{\Delta T_{Panel\;without\;PCM}}$$

ΔT_Panel with PCM_ is the amplitude of the temperature output curve at the specimen surface on the cold side where the experimental thermocouples were placed. This value was calculated for the mass fraction of PCM equal to 1.8%, 5% and 10%. ΔT_Panel without PCM_ is the amplitude of the temperature output curve of the specimen surface corresponding to 0% PCM (reference case). The ratio is a magnitude that quantifies the performance of the PCM and if equal or near to 1 this meant that PCM was not operating (charging and discharging). The optimum working conditions of PCM correspond to when the decrement factor is at a minimum. The numerical model was the same as that used in the validation phase of the model. The differences were the following: (i) the room temperature was constant, (ii) the PCM fraction varied, and (iii) the input temperature curve was a trapezoidal temperature wave of period equal to 1 day and defined by steps of 6-h duration. The point matrix was formed by the mean temperature and the amplitude, as shown in Figure 10. Calculations were performed automatically using a Python Programming Language script integrated into Ansys Workbench developed by the authors.

The results were represented as temperature contour plots, as shown in Figure 11. The contour images, constructed with 240 thermal transient calculations, show the effect of PCM through the created indicator—decrement factor (between 0 and 1). The efficiency of the panel could improve by nearly 50%, depending on the input boundary conditions. There was a temperature region near the peak melting temperature of the PCM where the performance of the panel was at a maximum. The panel efficiency and the optimum region size increased with increase of the PCM fraction. The optimum amplitude of the input temperature also varied with the PCM content and greater values allowed the storage and release of higher amounts of energy leading to lower decremental factors. The thermal behaviour was smoothed out by the PCM melting and solidifying process. The behaviour was at an optimum when the input mean temperature was near 20 °C for a room temperature of 18–20 °C. 

## 4. Conclusions

This work presents and discusses the outcomes of an experimental campaign and parametric numerical simulation developed using a novel multifunctional facade panel integrated with phase change material (PCM). The thermal characteristics of the panel and its potential as a thermal regulator for indoor spaces were evaluated through laboratory testing using the hotbox heat flux meter method, as well as resourcing numerical simulations using Ansys Fluent software.

The experimental results showed that the equivalent thermal conductivity and thermal transmittance values for temperatures below and above the PCM phase change value increased as the temperature rose. However, during the PCM phase change transition, there was a decreasing tendency in the thermal characteristics as the temperature increased.

Comparing the experimental temperature results obtained for exterior and indoor scenarios, a small percentage of PCM was revealed to result in a thermal amplitude reduction.

A numerical model was developed to assess the impact of PCM incorporation into the panel under different boundary conditions. The simulations were developed using Ansys Fluent software and the models were validated by comparing the numerical results with the experimental data. A good agreement was achieved.

The panel showed the best performance when the mean external temperature and internal temperatures are close to the PCM melting peak temperature.

Based on the results of the parametric study, increasing the quantity of PCM leads to an improvement in overall thermal performance. In the tested temperature range, the thermal regulation capacity achieved through the melting and solidification process was enhanced.

Further numerical modelling should be developed to explore novel combinations of different PCMs with different melting temperatures. Additionally, a detailed analysis of the liquid fraction of the PCM should be undertaken.

## Figures and Tables

**Figure 1 polymers-15-02971-f001:**
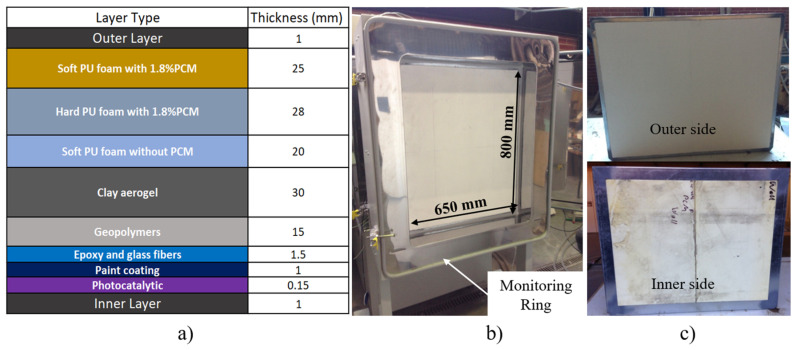
Multifunctional facade (**a**) layer layout constitution in mm, (**b**) dimensions of the tested panel and (**c**) outer and inner faces.

**Figure 2 polymers-15-02971-f002:**
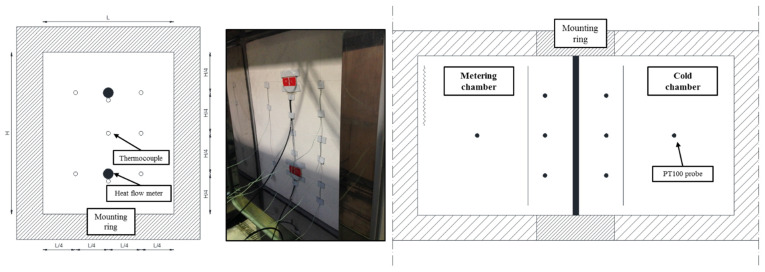
Test specimen instrumentation and surface sensor positioning.

**Figure 3 polymers-15-02971-f003:**
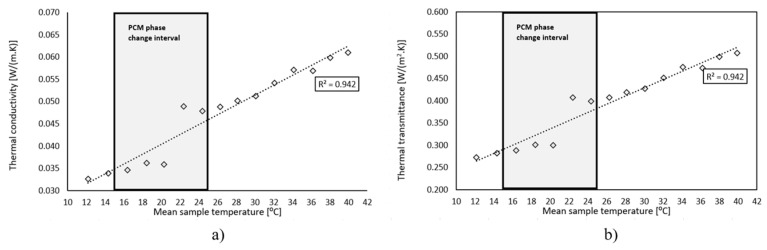
Thermal Conductivity (**a**) and Thermal Transmittance (**b**).

**Figure 4 polymers-15-02971-f004:**
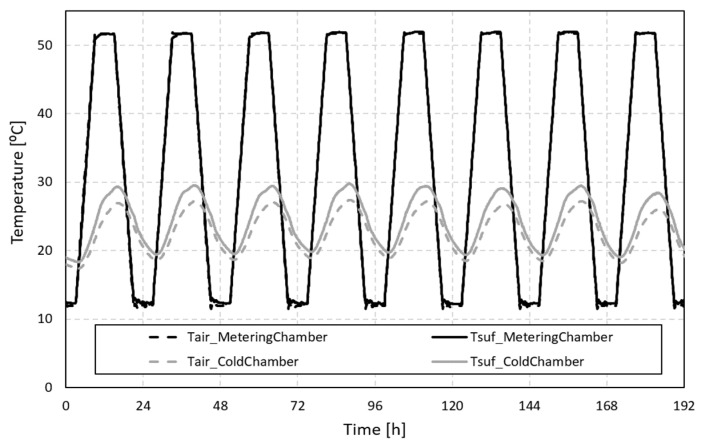
Temperature profiles for the 8-day cycle.

**Figure 5 polymers-15-02971-f005:**
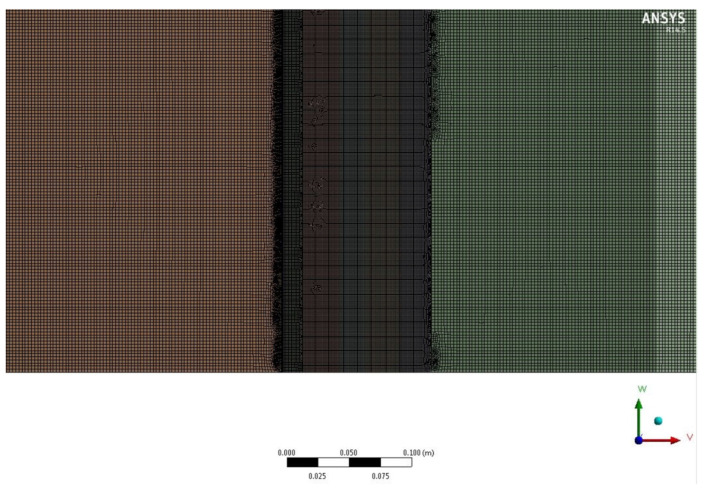
Numerical model mesh.

**Figure 6 polymers-15-02971-f006:**
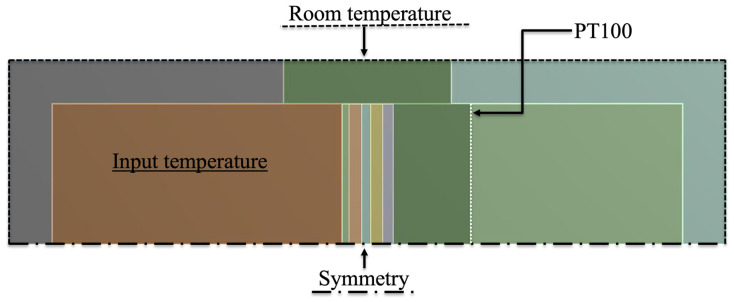
Numerical model and boundary conditions.

**Figure 7 polymers-15-02971-f007:**
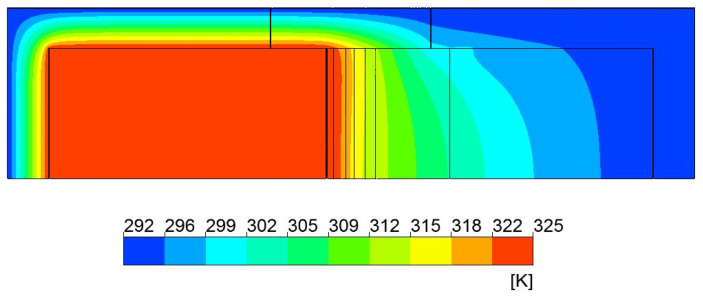
Temperature contours for maximum input temperature timestep.

**Figure 8 polymers-15-02971-f008:**
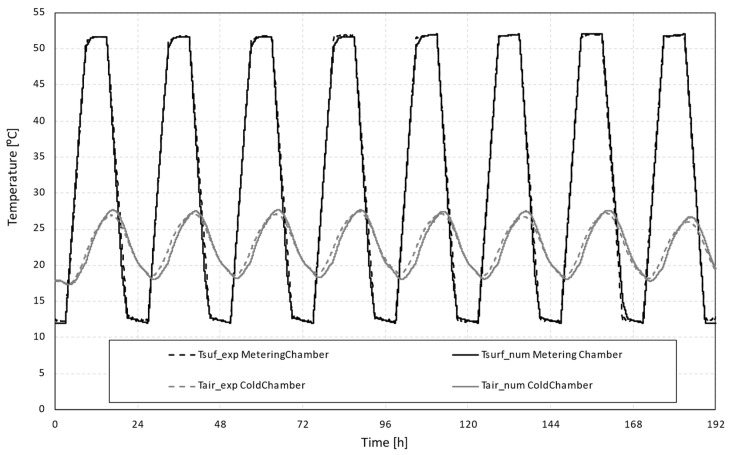
Comparison of experimental data and numerical simulation temperature profiles.

**Figure 9 polymers-15-02971-f009:**
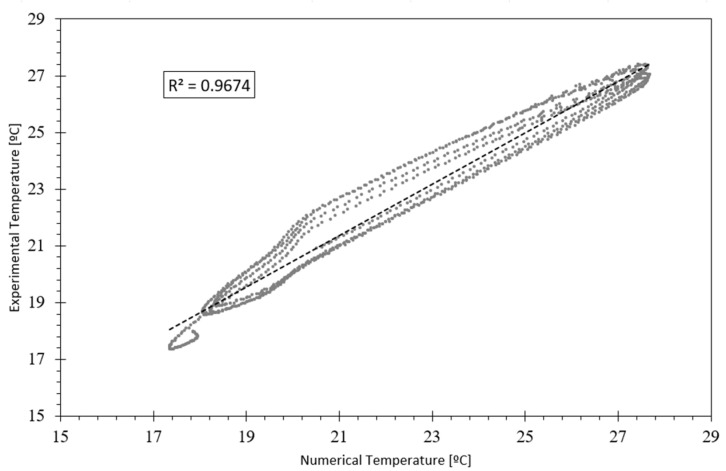
Experimental temperatures vs. numerical temperatures.

**Figure 10 polymers-15-02971-f010:**
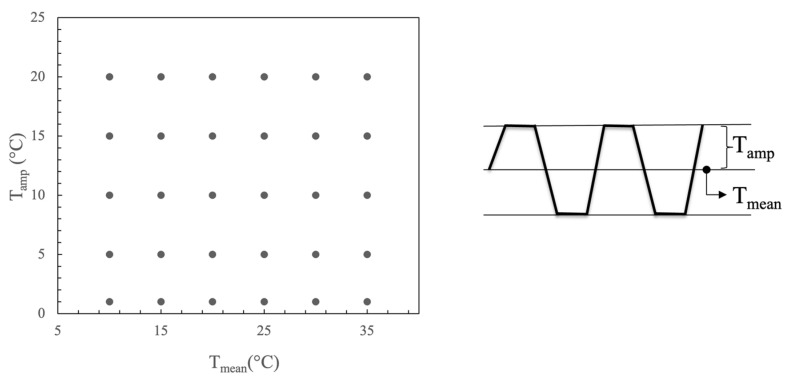
Mean and amplitude temperature input curves.

**Figure 11 polymers-15-02971-f011:**
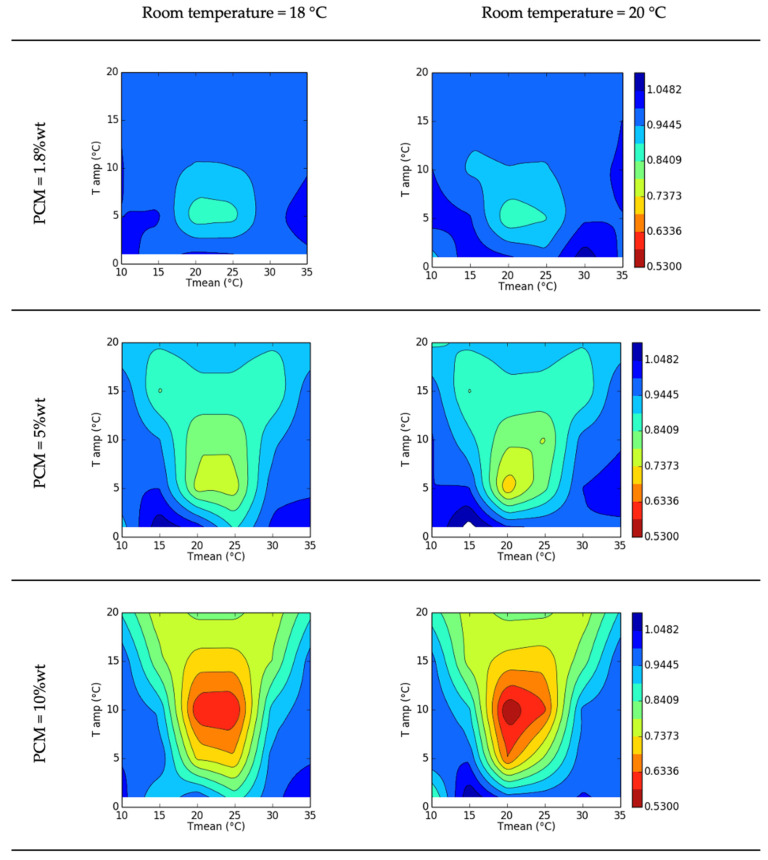
Decrement factor contour, f.

**Table 1 polymers-15-02971-t001:** Final properties of the multifunctional facade panel.

Panel Layers	Material	Thickness (mm)	Density (kg/m^3^)	Thermal Conductivity (W/m.k)	Specific Heat (J/kg.K)
SOFT PU FOAM LAYER	Soft PU foam with 1.8% PCM	25	101	0.037	-
HARD PU FOAM LAYER	Hard PU foam with 1.8% PCM	28	98	0.037	-
SOFT PU FOAM LAYER	Soft PU foam without PCM	20	101	0.037	1327
INSULATION LAYER	Clay aerogel	30	50	0.035	850
DURABLE LAYER	Geopolymers	15	1050	0.169	1000
EXTERNAL LAYER	Epoxy and glass fibres	1.5	1870	0.320	1500
INTUMESCENT LAYER	Paint coating	1.0	1500	0.200	1500
SURFACE COATING	Photocatalytic	0.15	1100	0.035	1500

**Table 2 polymers-15-02971-t002:** Properties of the PCM solely.

Density (kg/m^3^)	Melting			Thermal Conductivity (W/m.K)	
	Transition Temperature T_t,m_ (°C)	Melting Temperature T_m_ (°C)	Melting Latent Heat ΔH_m_ (J/g)	10 °C	20 °C
503	23.37	25.84	59.56	0.970	1.051

**Table 3 polymers-15-02971-t003:** Material Properties.

Chamber Material	Thickness (mm)	Density (kg/m^3^)	Thermal Conductivity (W/m.K)	Specific Heat (J/kg.K)	Viscosity (mm)
Galvanized steel	1.5	7833	54	465	-
Rockwool	125	70	0.0375	840	-
Zinc	1.5	7144	112.2	384.3	-
Interior air	-	7833	54	465	1.5

**Table 4 polymers-15-02971-t004:** Statistical indices.

Statistical Index	Panel
RMSE	0.68
CVRMSE	3.00
NMBE	−0.88
GOF	2.21

## Data Availability

Not applicable.

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
