# Peer review of "Thermal Performance of Multifunctional Facade Solution Containing Phase Change Materials: Experimental and Numerical Analysis"

_polymers, 2023, doi:10.3390/polym15132971_

Round 1

Reviewer 1 Report

The authors presented a numerical and experimental study on the Thermal performance of multifunctional facade solution containing phase change materials.

The main quantitative findings are to be presented in the abstract.

The novelty of the paper is to be clearly stated.

There some errors in the referencing (page 6,…) to be checked and corrected

Detail on the measurement techniques and data acquisition system are to be provided.

An experimental uncertainty study is to be performed.

For the numerical simulation, is it realistic to consider the problem as 2D? to be justified.

The solved governing equations are to be presented.

The boundary conditions are to be expressed mathematically.

Have you considered the effect of natural convection in the PCM model (after the melting).

A figure presenting the used mesh is to be presented.

A grid sensitivity test is to be performed.

What is the convergence criterion?

What is the time step?

Results related to PCM liquid fraction are to be presented and discussed.

Author Response

All responses are provided in the attached file. Thank you.

Reviewer 2 Report

The paper is interesting and can be published provided that the authors address the following minor comments:

1. Moderate English proofreding is required.

2. The novelty of the work should be highlighted.

3. The numerical method should be expalined in more details.

4. are the authors using a commercial software? if so, they should mention it.

5. Comparison with previously published work should be done.

6. The literature review can be enhanced by the following related works on PCM:

a) Investigation of using multi-layer PCMs in the tubular heat exchanger with periodic heat transfer boundary condition

b) Numerical investigation of rectangular thermal energy storage units with multiple phase change materials

c) Phase-change heat transfer of single/hybrid nanoparticles-enhanced phase-change materials over a heated horizontal cylinder confined in a square cavity

It needs moderate improvement.

Author Response

(The authors gave the same response as above.)

Reviewer 3 Report

Polymers-2394431 Review of “Thermal performance … numerical analysis By Amaral…Vicente   This paper compares the experimental measurements of thermal performance
and the numerical results using Fluent. The aim is to study the thermal
characteristics of facades including phase change materials (PCM).  The results show that the experimental and computed results are in good
agreement—as expected.  From a practical point of view, the agreement is expected. Is anything
gained from the paper and the description? Unfortunately no. The paper simply validates that the experiment and the analysis match well.
This is always assumed to be true for such a well defined
experiment/analysis.  I cannot recommend publishing this paper in the journal.
It really confirms what is expected.

Author Response

(The authors gave the same response as above.)

Reviewer 4 Report

In this study, the authors have conducted both experimental and numerical investigations to determine the thermal transmittance of Pristine as a phase change material (PCM). They have compared their experimental and numerical findings, utilizing the finite element method to solve the problem numerically. The manuscript exhibits an adequate technical structure. However, to enhance its suitability for publication in the reference journal, the authors should address the following comments:

1- Please expand the introduction section by incorporating reviews of papers published in the most recent years, 2022 and 2023.

2- For equations that are not original contributions from the authors, include relevant references.

3-  Ensure that equations are numbered sequentially and correctly.

4-  In Figure 2, the text is not easily readable. Please present the figure at a larger scale for improved legibility.

 5- On page 6, following Figure 2, the sentence "Error! Reference source not found. presents the sensors…" contains a typographical error. Please correct this and any other typographical errors throughout the manuscript.

6- Since the authors have employed the finite element method in their work, it is advisable to conduct a "grid independence process" to ensure the reliability of the numerical simulations. Please consider implementing this process.

7- In Table 2, which presents the thermophysical properties of the chamber walls, include the relevant references.

8- To improve the clarity of the results in Figure 7, use different line patterns or symbols in addition to color variations to distinguish between the data.

9- For Figure 10, remove the borders and clarify whether the color legend applies to both figures in a row, or if the color legend for the left side has been unintentionally omitted.

By addressing these comments, the manuscript can be significantly improved, increasing its potential for publication in the reference journal.

The text readability should be improved, and some mistakes should be corrected. For example, see below:

    "Sovetova et al. [15] presented a numerical analysis to evaluate the impact of the use of a microencapsulated PCM layer right between the outer layer (cement plaster and ceramic tile) and the concrete layer on the façade of a building in the hot desert region."

    Mistake: The phrase "right between" is informal.

    Correction: Replace "right between" with "located between".

    "Bahrar et al [18] performed a multiscale experimental characterization textile reinforced concrete panels with microencapsulated PCM, using the hot box method, as well as the development of a numerical model to accurately reproduce the thermal performance of the building envelope."

    Mistake: The sentence lacks proper punctuation and is a run-on sentence.

    Correction: Break the sentence into two: "Bahrar et al. [18] performed a multiscale experimental characterization of textile-reinforced concrete panels with microencapsulated PCM using the hot box method. They also developed a numerical model to accurately reproduce the thermal performance of the building envelope."

    "Li et al.[19] proceeded to assess a thermos-economic and environmental analysis of PCM embedded walls in rural residence of Northeast China, recurring to EnergyPlusTM."

    Mistake: The term "thermos-economic" is incorrect.

    Correction: Replace "thermos-economic" with "thermo-economic".

    Mistake: The phrase "recurring to" is not appropriate in this context.

    Correction: Replace "recurring to" with "using".

    "In a former study, Li et al [20] explored both the thermal and optical behaviour of a non- ventilated multilayer glazing facade filled with PCM for severe cold climatic conditions."

    Mistake: The phrase "non- ventilated" has an unnecessary space.

    Correction: Remove the space, making it "non-ventilated".

    "The multifunctional panel developed in the scope of this research has many interesting features, scoping from the use of PCM, recycled materials, and innovative material processing techniques to the auto-cleansing function of the exterior surface."

    Mistake: The word "scoping" is not suitable in this context.

    Correction: Replace "scoping" with "ranging".

Author Response

(The authors gave the same response as above.)

Round 2

Reviewer 1 Report

Accept in present form

Author Response

Accepted in present form.

English language fine. No issues detected.